# Exploring Genetics by Environment Interactions in Some Rice Genotypes across Varied Environmental Conditions

**DOI:** 10.3390/plants13010074

**Published:** 2023-12-26

**Authors:** Mohamed I. Ghazy, Mohamed Abdelrahman, Roshdy Y. El-Agoury, Tamer M. El-hefnawy, Sabry A. EL-Naem, Elhousini M. Daher, Medhat Rehan

**Affiliations:** 1Rice Research and Training Department, Field Crops Research Institute, Agricultural Research Center, Kafrelsheikh 33717, Egypt; m_ghazy2050@yahoo.com (M.I.G.); roshdyrrtc@gmail.com (R.Y.E.-A.); tito.307@gmail.com (T.M.E.-h.); sabryelnaem@gmail.com (S.A.E.-N.); elhousani.daher@yahoo.com (E.M.D.); 2Department of Plant Production and Protection, College of Agriculture and Veterinary Medicine, Qassim University, Buraydah 51452, Saudi Arabia; 3Department of Genetics, Faculty of Agriculture, Kafrelsheikh University, Kafr El-Sheikh 33516, Egypt

**Keywords:** multilocation, climate change, heat stress, GGE biplot, AMMI analysis, yield stability, genotypes ranking, sustainability, biodiversity

## Abstract

Rice production faces challenges related to diverse climate change processes. Heat stress combined with low humidity, water scarcity, and salinity are the foremost threats in its cultivation. The present investigation aimed at identifying the most resilient rice genotypes with yield stability to cope with the current waves of climate change. A total of 34 rice genotypes were exposed to multilocation trials. These locations had different environmental conditions, mainly normal, heat stress with low humidity, and salinity-affected soils. The genotypes were assessed for their yield stability under these conditions. The newly developed metan package of R-studio was employed to perform additive main effects and multiplicative interactions modelling and genotype-by-environment modelling. The results indicated that there were highly significant differences among the tested genotypes and environments. The main effects of the environments accounted for the largest portion of the total yield sum of squared deviations, while different sets of genotypes showed good performance in different environments. AMMI1 and GGE biplots confirmed that Giza179 was the highest-yielding genotype, whereas Giza178 was considered the most-adopted and highest-yielding genotype across environments. These findings were further confirmed by the which–won–where analysis, which explained that Giza178 has the greatest adaptability to the different climatic conditions under study. While Giza179 was the best under normal environments, N22 recorded the uppermost values under heat stress coupled with low humidity, and GZ1968-S-5-4 manifested superior performance regarding salinity-affected soils. Giza 177 was implicated regarding harsh environments. The mean vs. stability-based rankings indicated that the highest-ranked genotypes were Giza179 > Giza178 > IET1444 > IR65600-77 > GZ1968-S-5-4 > N22 > IR11L236 > IR12G3213. Among them, Giza178, IR65600-77, and IR12G3213 were the most stable genotypes. Furthermore, these results were confirmed by cluster-analysis-based stability indices. A significant and positive correlation was detected between the overall yield under all the environments with panicle length, number of panicles per plant, and thousand grain weight. Our study sheds light on the notion that the Indica/Japonica and Indica types have greater stability potential over the Japonica ones, as well as the potential utilization of genotypes with wide adaptability, stability, and high yield, such as Giza178, in the breeding programs for climate change resilience in rice.

## 1. Introduction

Rice (*Oryza sativa* L.) plays a vital role as a staple component in the diets of significant portions of the global population. As the world’s population is projected to reach 10 billion by 2058 [1], the importance of rice production becomes even more critical. Approximately 755 million tons of paddy rice are produced annually from about 162 million hectares of land [2]. However, in order to ensure food security and combat poverty, there is a pressing need for a substantial increase, exceeding 60%, in high-quality rice production [3,4].

To meet this escalating demand, programs of rice breeding have implemented various strategies to enhance yield potential and develop high-yielding rice varieties. These strategies have often focused on utilizing specific germplasms tailored to a single environmental condition, such as submergence, upland, lowland, salinity, or drought. However, the ever-growing threat of climate change poses new challenges, emphasizing the importance of stability and adaptability in rice cultivars across different environmental conditions. Therefore, it is crucial to model genotype-by-environment interactions (GEI), quantify genotypic resilience, and assess the stability of the parental genotypes that are frequently used as donor varieties. Such an approach is necessary to increase the efficiency in genotype selection and determine the adaptability of the genotype under multienvironment trials (METs) [3,5,6,7,8,9]. Parental selection for crosses might consider high adaptation (genotype capability to positively adapt to surrounding stressors) and yield stability (genotype capacity to react in relation to the environment’s yield prospective) across various environments (places, seasons, or both). Taking these considerations into account, parent selection is also critical for reproduction tasks looking for a greater area of protection, particularly in places with diverse environmental and soil conditions [10]. Wide adaptability and stability under different environments are the most important characteristics that should be imbedded as criteria for varietal selection in breeding programs. Rice genotypes with widespread adaptability could be recommended as elite parents in rice breeding programs to improve general adaptability in various climates, particularly with the unpredictability of climate change events and the potential occurrence of extreme stressors simultaneously or successively. Several statistical approaches have been created to enhance the precision of genotype x environment interactions and to aid understanding the stability as well as adaptability of tested genotypes [3,11].

Grain yield is considered the most reliable determinant of genotypic performance, mainly determined by the additive effects of genotype (G), environment (E), and the nonadditive effect of GEI [12]. Evaluating different rice varieties across diverse rice production locations provides essential insights into their performance and resilience to the inherent environmental factors unique to each location. Furthermore, this approach allows for tailored genotype selection tuned to distinct natural environments [13]. Additionally, such multilocation preliminary yield trials facilitate the selection of promising varieties by demonstrating their yield potential and stability across diverse environments [14]. So long as the environment differs in terms of climatic conditions, the varieties that exhibit stability across these environments demonstrate greater resilience to the effects of climate change.

Several statistical methods have been developed to measure GEI, including regression coefficient [15], sum of squared deviations due to regression [16], and additive main effects and multiplicative interaction (AMMI) [17]. Another method, known as G + G × E (genotype + genotype × environment, GGE) polygon view, incorporates a graphical approach (GGE biplot), which characterizes the mega-environment, ranks varieties, and identifies the best genotypes in each environment (which–won–where) [18,19].

In recent a development, a novel R package named “metan” has been specifically designed to analyze multienvironment trials (METs) [20]. This package offers a workflow-based procedure that encompasses sequential functions for assessing commonly used parametric and nonparametric stability statistics [21]. The metan package provides a comprehensive set of tools for MET data management, manipulating, analyzing, and visualizing the MET data. It has been successfully employed to quantify yield stability in various crops, including rice [14], lentil [22], wheat [23], soybean [24,25], sugarcane [26], and others [27,28,29,30,31].

In the current investigation, the yield performance of different rice genotypes was evaluated at different rice growing environments. These genotypes are commonly employed as donors for different breeding purposes, such as salinity, water shortage, and disease resistance. These genotypes are tested under environments where heat stress, low humidity, and salinity stress are naturally inherited in these locations. This study aims to conduct a comprehensive multilocation evaluation of yield performance, identifying genotypes with both high yield potential and stability in diverse locations.

## 2. Results

### 2.1. GY Combined AMMI Analysis of Variance and Genotypic Variability

To cope with climate change, 34 genotypes were assessed at four locations during two seasons of study (2021 and 2022). The obtained results indicated highly significant differences among the genotypes across the different environments under study (Figure 1A,B and Table 1). The AMMI model analysis of variance for the 34 genotypes across the four locations throughout the two seasons exhibited a significant influence of all the factors analyzed on the genotypes’ yield performance (Table 1). The main effects of environments accounted for a substantial portion, as much as 71.29%, of the total sum of the squared deviations for the genotypes’ yield, while the proportion that accounted for G and GEI was 14.77 and 13.94, respectively. Accordingly, 29.32% of the grain yield variability could be attributed to identifying genotypes with narrow adaptability.

The variability among the genotypes is supported by the differences between the mean yield values of the genotypes under study as they ranged from 238.5 g m^−2^ for Giza177 (G1) in 2022 at the Alexandria location to 1209 g m^−2^ for Giza179 (G3) in 2022 at the Gemmiza location (Figure 1A, Appendix A). Furthermore, the average yield production throughout the locations was also highly changeable. The average yield at Alexandria revealed the lowest values (525.9 g m^−2^ and 525.5 g m^−2^ for the two seasons, respectively), while the Gemmiza site recorded the maximum values (987.7 and 990.4 g m^−2^, respectively) for the two seasons of study (Figure 1B).

### 2.2. GEI-Structure-Based Additive Main Effect and Multiplicative Interaction Model

The decomposition of the effect in the interaction-related multiplicative variance of the GEI was depicted by AMMI analysis and yielded four significant principal components of interaction (IPCAs, Table 1). These four IPCAs explained 99.5 percent of the total GEI effects, contributing 88.9%, 7.9%, 2.2%, and 0.5% for IPCA1, IPCA 2, IPCA3, and IPCA4, respectively. Meanwhile, IPCA5, IPCA6, and IPCA7 displayed nonsignificant impacts, and their ratio amounted to 0.5. IPCA1 and IPCA2 together gathered about 96.9%, explaining that both IPCAs are the best predictive model and sufficient for explaining the GEI. These results further indicate the extensive interaction of multivariate datasets with the PCA output, subsequently extracting and understanding the patterns regarding GEI that existed in the yield performance of the 34 genotypes under the different environments.

**Table 1 plants-13-00074-t001:** AMMI analysis of variance model and GEI decomposition.

Source	Df	Sum Sq	Mean Sq	F Value	Pr (>F)	Proportion	Accumulated
ENV	7	32,277,639	4,611,091	5554.827	0		
REP(ENV)	16	13,281.69	830.1054	1.703935	0.042241		
GEN	33	6,685,134	202,579.8	415.83	0		
GEN:ENV	231	6,311,279	27,321.56	56.08221	8.8 × 10^−275^		
PC1	39	5,612,970	143,922.3	295.43	0	88.9	88.9
PC2	37	499,545.4	13,501.23	27.71	0	7.9	96.9
PC3	35	141,380.5	4039.443	8.29	0	2.2	99.1
PC4	33	28,625.39	867.4361	1.78	0.0054	0.5	99.5
PC5	31	15,441.88	498.1253	1.02	0.439	0.2	99.8
PC6	29	10,612.79	365.9583	0.75	0.8262	0.2	100
PC7	27	2703	100.1111	0.21	1	0	100
Residuals	528	257,225.6	487.1697				
Total	1046	51,855,839	49,575.37				

ENV: environments; REP: replicates; GEN: genotypes; PC: principal components; Df: degree of freedom; Sq: squares; and Pr: probability.

### 2.3. Detection of Stable and High-Yielding Genotypes via AMMIs Model Biplot

To identify high-yielding and stable genotypes across the environments, the AMMI1 biplot was generated. The AMMI1 biplot plots the means of the genotypes and environments against their PCA1 (Figure 2A). The AMMI1 biplot revealed that those environments located at Alexandria (E3 and E4) and Kharga oasis (E7 and E8) produced the lowest yields across the genotypes. Similarly, Sakha103 (G32) showed the minimized grain yield average across the different environments presented in the current research (Figure 2A and Appendix A). Furthermore, the AMMI1 biplot exhibited that Giza179 (G3) had the uppermost yielding values, whereas Giza178 (G2) was considered the most widely adopted and highest-yielding genotype across the environments. Based on the genotypes’ performance with PC1 estimates, a value close to zero means that the genotype has general adaptability under the different environments. Morobereccan (G17) and Nerica 7 (G16) were the most stable genotypes, owing the least PC1 values of 0.22 and 0.3, respectively (Figure 2A and Appendix A).

The AMMI2 biplot pinpointed the genotype stability performance regarding yield based on IPCA1 and IPCA2 values (Figure 2B). The genotypes and environments with lower IPCA1 and IPCA2 values that are plotted in close proximity to the plot origin are considered the most stable ones. This explains the reduced interaction ability of genotypes and their adaptability regarding the different environments under study. Based on the AMMI2 biplot, Nerica 7 (G16), Morobereccan (G17), IR11L236 (G27), and Giza179 (G3) are demonstrated to be considerable genotypes. Both the AMMI1 and AMMI2 biplots indicated the instability of Giza177 (G1) and Vandana (G30) for the different environments in the present research.

### 2.4. Which–Won–Where Approach Based on GGE Biplot for Detecting the Best-Performing Genotypes

The GGE biplot polygon view based on the which–won–where structure of a MET approach is the simplest and most efficient method for detecting the genotype and its environmental interaction (Figure 3A). It is used for interpreting GEI and detecting superior genotypes across different environments. In the current investigation, the biplot polygon view showed the tested environments in two different sectors. The environments of the locations Gemmiza (E5 and E6) and Sakha (E1 and E2) were located in the same sector, while the other two locations (Alexandria (E3 and E4) and Kharga oasis (E7 and E8)) were separated in another sector. The genotypes at the corner of each section of those environments had the highest yield for the corresponding environments. The genotypes joined by the polygon are the farthest from the origin and called vertex genotypes. Those vertex genotypes located in the same sector or close to a specific environment are considered the best genotypes for this environment. Accordingly, Giza179 (G3) is the best-performing genotype in Sakha and Gemmiza, followed by Sakha Super 300 (G11). Likewise, N22 (G21) followed by IET1444 (G15) are considered the most suitable genotypes for environments E7 and E8 of Kharga oasis. IET1444 (G15) and Giza178 (G2) had stable performance for the environments located in their sectors. Furthermore, Giza178 (G2) exists in the middle of all the environments; consequently, it is stable across all the study environments. In contrast, Giza177 (G1) is situated on the opposite side of E3, E4, E7, and E8, which reflects the minimum appropriateness for these environments. Similarly, Dular (G31) exists on the opposite side of E1, E2, E5, and E6, reflecting the same situation.

### 2.5. Grain Yield Versus Weighted Average of Absolute Score Stability Index Biplot

Various stability statistics were estimated and presented in the Appendix A. Based on GY, which considers the main breeder selection criteria, genotypes Giza179, Giza178, IET1444, IR65600-77, GZ1968-S-5-4, and N22 achieved the highest rank in this regard. The weighted average of absolute scores (WAASB) was also used to better identify the most adapted genotypes across the different environments based on the mean GY and stability. The GY× WAAS biplot displayed the distribution of the tested rice genotypes and environments based on the genotypes’ GY mean and WAASB values, as presented in Figure 3B.

The first quadrant I, contains those genotypes that are low yielding and unstable across all tested environments. Among these genotypes Giza177 (G1), G4, G5, G7, G8, G19, G30, and G31. These genotypes are less desirable as they showed lower grain yield and inconsistent performance compared to the mean of overall grain yield. In addition, this quadrant has low GY environments E3, E4, E7 and E8. Accordingly, the genotypes and environments located in this quadrant have the largest response to GEI. The second section (quadrant II), contains the environments E1, E2, E5 and E6, coupled with the genotypes that have a GY above average with a high GEI response, such as G6, G10, G11, G34, and G21. These genotypes are less reliable as they have high yield under normal conditions, but didn’t display consistently under harsh environments. GY × WAAS biplot also presented genotypes that showed relative stable performance across the evaluated environments. These genotypes were existed in the quadrants III and IV. G13, G14, G16, G17, G20, G23, G26, G29 and G32 presented minimum yield but stable performance across the tested environments (Figure 3B). Those genotypes are more suitable for harsh environments. At the same time, G2, G3, G9, G12, G15, G18, G22, G24, G27, G28 and G33 located in quadrant IV. This quadrant contains the genotypes showing high yield performance without being influenced significantly by specific environmental conditions. These genotypes are desirable for broad adaption ability.

### 2.6. GGE Biplot—Means Versus Stability Model and Ranking of Rice Genotypes’ Performance

The genotypes’ mean performance versus stability biplot provides a visual tool for discriminating the tested genotypes (Figure 4A,B). This biplot presents the two PCs (1 and 2), which, for their additive percentage, explain the G + GE effects, respectively. The single-arrowed line in the biplot (Figure 4A) is the average environmental average (AEA) and points towards higher mean performance across the tested genotypes. Regarding the AEA, the average environment corresponding to the average values of the two PCs is pinpointed by the arrowhead in Figure 4A and further circled in Figure 4B. The genotypes located in the circle are considered to be the best genotypes. The perpendicular line to the AEA is called the average ordinate environment (AOE), and the intersection is the point that represents both the average mean performance and high stability. Other perpendicular lines linking the genotypes to the AEA explain the stability of the genotype. The closeness of genotypes to the AEA explains their stability across environments. However, using the ranking of the biplot, the ideal genotype is Giza178 (G2), being in the center of the circle. Giza178 had a high yield and the best adaptability among the other genotypes under consideration. Giza179 (G3) also exhibited high yielding ability. Giza 177 (G1) manifested a decrease in yielding ability with decreased stability compared to the other genotypes under study. The ranking of the genotypes from the worst to the best could be followed using the AEA direction. The yield average reduced in Sakha103 (G32) to the lowest values across the environments, followed by Giza177 (G1), Morberekan (G17), Dular (G31), IRAT112 (G20), and Nerica 9 (G14), following the AEA direction until Giza179 (G3). The uppermost-ranked genotypes were assigned for Giza179 (G3) > Giza178 (G2) > IET1444 (G15) > IR65600-77 (G22) > GZ1968-S-5-4 (G18) > N22 (G21) > IR11L236 (G27) > IR12G3213 (G24).

### 2.7. Cluster-Analysis-Based Stability Indices

Several stability indices were measured based on the genotypes’ yield records in the environments under the present constructed study. Stability indices such as environmental variance, mean variance component, GE variance component, joint regression analysis, Tai’s stability statistics, coefficient of variance, superiority index, and AMMI-based stability statistics are normally used to rank the genotypes based on their estimated values. These indices are Shukla_R, Wi_g_R, Wi_f_R, Wi_u_R, Ecoval_R, Sij_R, Pi_a_R, Pi_f_R, Pi_u_R, Gai_R, S1_R, S2_R, S3_R, S6_R, N1_R; Appendix A. The indices’ values corresponding to the tested genotypes were manipulated based on the squared Euclidean distance to conduct hierarchical cluster analysis via Ward’s method (Figure 5). This method was used to group the genotypes with the same stability in the same cluster. Based on this fact, the tested rice genotypes were grouped in two main clusters (CL), resembling the main grouping pattern. Each CL was divided into two subclusters (SCL). The first SCL colored with red contains genotypes that had similar stability measures. Those genotypes (Giza179, Giza178, IR65600-77, IR11L236, IR12G3213, and IR12G3222) have the highest overall mean performance across the different environments. These genotypes are among the top-ranked ones based on mean vs. stability ranking. Furthermore, IET1444 was clustered with N22, and they are the top-performing genotypes under Kharga oasis environments. GZ1968-S-5-4 is the top-performing genotype under Alexandria conditions, whereas IET1444 is considered among the top-four-performing genotypes. For the Gemmiza location, Sakha104 and Sakha 107 are among the top-performing genotypes, whereas Sakha104 and Sakha109 are categorized at the same rank in the overall mean performance of the genotypes as compared to other genotypes. These results confirm the goodness of the stability indices in ranking the genotypes and their utilization in clustering the genotypes into different groups.

The Sakha Super 300, Sakha101, and Sakha 108 genotypes showed top performance in the Sakha and Gemmiza locations. However, their rank deteriorated in the unfavorable environments of Alexandria and Kharga oasis as compared to the other tested genotypes. Giza177, Sakha105, Sakha102, and Saka 106 were grouped together in the middle subcluster, indicating their average performance under favorable conditions and low performance under unfavorable conditions. At the same time, IRAT17, Giza182, Azucena, and Vandana were grouped together and provided a slightly more stable performance in unfavorable conditions and an unstable performance under favorable conditions when compared to the other genotypes. Moreover, IRAT112, IR69116, IR6500-127, and Dular in the same subcluster had average performance under harsh environments and low performance under favored environments.

The Sakha 103, Nerica 9, and Moroberekan genotypes grouped together in the purple subcluster and displayed a reduction in their performance with regard to the overall performance under the different environments. In the last subcluster, IR69432, A22, and Nerica7 located together due to their average ranking across the different environments.

### 2.8. Correlation Coefficient Analysis

The correlations among the studied yield-related characteristics and yield were analyzed under the different environments (Figure 6). All the evaluated characteristics have a significant correlation with yield. A significant and positive correlation was detected between the overall yield under all the environments with panicle length (r= 0.825), number of panicles per plant (r = 0.904), and thousand grain weight (r = 0.491). In contrast, the total yield performance negatively correlated with the genotypes’ sterility percentage under all the environments. Interestingly, panicle length showed a highly significant and positive correlation with yield under harsh conditions in contrast to the negative ones under favored environments. Furthermore, sterility was negatively correlated with all the measured characteristics. In the density plots presented in the diagonal, it is noted that E1, E2, E6, E8, and E2 have the highest peaks compared to the others for yield, panicle length, number of panicles per plant, thousand grain weight, and sterility percentages, respectively.

## 3. Discussion

Breeding for yield stability across different environments is becoming the most popular program with the current waves of climatic changes. Annually, there are major fluctuations in climate records, especially with regard to water scarcity, temperature, and humidity. Those stressors are considered among the main ones affecting rice production [10]. Identifying the most suitable genotypes for a specific climatic location involves an important step of starting a breeding program for such an environment. AMMI analysis of variance of 34 rice genotypes over eight environments for grain yield is provided in Table 1. Significant differences were detected among E, G, and GEI. This means that the genotypes showed different behavior in their environments. This enables the breeder to justify the selection of genotypes based on the magnitude of interaction with the environment [32,33]. Environments accounted for the highest sum of squares, 70.68%, explaining the highest differences among them. This indicates that different sets of genotypes appeared to be high-yielding in different environments. Accordingly, selection of the genotypes based on the environment is effective while considering genotype performance and GEI. The high effect of environments was mainly due to the unique environmental features for each location. These variability among the environmental conditions may have activated some yield-enhancing genes in different genotypes, which developed a significant GEI and resulted in high yield variability (Figure 1A,B). While investigating the genotypic yield stability across environments, the research reported by Barona et al. 2021, Dewi et al. 2014, Tariku et al. 2013, and Zewdu et al. 2020 [34,35,36,37] also concluded that the environmental effect recorded the highest impact. Meanwhile, other investigations reported genotypes’ effects to be the most effective impacts. In these investigations, the differences between environments were mainly due to one effect, such as identifying the most stable rice genotype across different fertilizer levels [14].

Apparently, our investigation confirms that high variability exists between the tested environments. Two locations had favored environments, Sakha and Gemmiza (Appendix A), where the environmental conditions and soil properties are suitable for good rice growth. Contrarily, Kharga oasis and Sobahya had unfavored environments. Kharga oasis has high temperature records during the growing seasons together with low humidity, while the Alexandria site suffered from poor soil conditions. These conditions were reflected in the genotypes’ yield performance across environments (Figure 1A,B). Consequently, the rankings of the tested genotypes, based on yield performance, were subject to changes in specific locations. Giza179, Sakha Super 300, Sakha 108, and Giza178 developed the highest grain yield under the favored conditions of Sakha and Gemmiza. Meanwhile, GZ1968-S-5-4, IRAT170, IET1444, and Giza179 were the best at the Alexandria location despite affecting the soil with salinity. Moreover, at the unfavored environment of Kharga oasis, with heat stress accompanied by low humidity conditions, genotypes N22, IET144, Giza178, and IR65600-77 recorded the best values and the highest ranking. These findings demonstrate the influence of GEI on the performance of the tested genotypes. GEI reduced the performance of the evaluated genotypes by affecting their yield under unfavored conditions. This output is supported by the findings of others [38,39].

To quantify the magnitude of the GEI in the current investigation, the GEI principal components (PCs) were assessed. GEI was partitioned into seven PCs (Table 1). The first two significant IPCs (PC 1 and PC 2) accounted for approximately 96.9% of the interaction between the E and G. This indicates that these two PCAs were sufficient to present the GEI complex patterns. Consequently, all the interaction information could be speculated by plotting these two PCs. Our findings provide proofing for the results of several earlier investigations obtained by others that support the idea that the first two IPCs of the AMMI model are the most important during quantifying GEI [14,34,35,36,37].

The AMMI model provides an effective analytical procedure to understand the GEI through a graphical biplot tool to clearly identify the ideal genotypes [21,40,41]. Among these are AMMI1 and two biplots. AMMI1 revealed that Giza 178 is the best genotype with regard to stability and mean performance across environments (Figure 2A). Genotypes with small interaction with the environment have IPCA1 values close to zero [5,21]. Consequently, Giza178 showed wider adaption to the environments (normal, heat stress, and salinity stress) under the current research, and high yielding performance. Contrarily, Giza177 showed small adaptability to the tested environments. This is mainly due to lacking specific genes that could provide the genotype resistance to the harsh environments of Kharga oasis and Sobahya. The AMMI2 biplot summarizes information based on the first two PCs (PC1 and PC2 in GEI). Genotypes with low IPCA1 and IPCA2 values have low interaction with the environments [38,39,40,42,43,44]. These genotypes are centered near to the origin of the AMMI2 biplot. Among these genotypes, Giza179 had high yielding performance and had general adaptability to the conditions under study. Giza179 is an Egyptian rice cultivar bred for climate change resistance [9,45].

Evaluating the genotypes across multilocation trials provide the fundamentals regarding variation. The GGE biplot clarifies the best-performing genotypes across all the environments [42,46]. The polygon view of the GGE biplot (which–won–where) illustrated the winning genotypes for each environment under study (Figure 3A). This biplot view considers the most efficient and simplest method for characterizing the genotypes and their interaction with the environment [5,14]. In this biplot, the genotypes with the longest distance from the biplot origin in the same direction or close to one or more from the environment are considered the best genotypes for those environments. Giza179 is the superior-performing genotype in the Sakha and Gemmiza environments, followed by Sakha Super 300. These environments are considered ideal for rice crop growth in Egypt. Meanwhile, in a harsh environment where heat stress is excited in Kharga oasis, N22 and IET1444 were preferable. This finding is supported by N22 and IET1444 harboring specific genes for heat and water shortage stress tolerance [43,44,47,48,49]. Furthermore, IET1444 and Giza178 were located in the middle between the eight studied environments regarding their stable performance [5], while Giza177 was inferior for the environments located in Kharga oasis and Sobahya. This was mainly due to the harsh environments coupled with heat and salinity stress. Giza177 is a popular Egyptian variety with high grain quality characteristics but is sensitive to heat and salinity stress [47,50].

We further calculated the stability index of WAAS to consider the sum of the absolute values of the genotypes’ IPCAs [39,43]. Then, a GY × WAAS biplot was generated to better clarify the superior-performing genotypes. In terms of WAAS values, Moroberekan represents the minimal and desirable WAAS value. Moroberekan is a drought-blast-resistant genotype but with poor yield potential under favored conditions [47,51]. These features provided the genotype with stability performance across the different environments tested in the current study. However, as the biplot considers both GY and WAAS values for the genotypes, Giza178 and Giza179 are the superior ones (Figure 3B). These genotypes had high yield and minimum estimates for WAAS stability index. The GY × WAAS biplot further divided the tested genotypes into four different categories (i.e., high-yielding stable, high-yielding not stable, low-yielding stable, and low-yielding unstable genotypes). This classification provides a clear vision for rice breeders to predict the genotypic performance of each genotype under study prior to integrating them in the different programs.

It is essential to assess the average stability together with the genotype’s performance through the means versus stability model of a GGE biplot (Figure 4A,B). This model is capable of detecting high-ranking genotypes with great stability performance based on AEC decisions [23,52]. Giza179 is considered the highest, whereas the ideal genotype was Giza178 via being high-yielding and more stable than Giza179. In contrast, Giza177 had low stability with sensitivity to harsh environments under study, which decreased its overall yield performance. Based on AEC ranking, the lowest-ranked genotypes are Sakha103, Giza177, Moroberekan, Dular, IRAT112, and Nerica 9, while the top-ranked genotypes are Giza179 (G3) > Giza178 (G2) > IET1444 (G15) > IR65600-77 (G22) > GZ1968-S-5-4 (G18) > N22 (G21) > IR11L236 (G27) > IR12G3213 (G24). Among them, Giza178, IR65600-77, and IR12G3213 were the most stable ones. These genotypes could be utilized in breeding programs while having high yielding ability under unfavored environments and also maintaining good performance under normal conditions.

To further identify the groups of the genotypes based on the different stability indices, cluster analysis was conducted. It is worth mentioning that, based on this analysis, the genotypes with the highest overall mean performance across the different environments were clustered in a single SCL. These genotypes are Giza179, Giza178, IR65600-77, IR11L3213, IR12G3213, and IR12G3222, which are the top-ranked ones based on mean vs. stability ranking (Figure 5).

Correlation coefficients indicated that number of panicles per plant exhibited the highest positive correlation with grain yield under all the environments. However, the overall correlation coefficient between panicle length and grain yield records was significantly positive, and the coefficient was positive under unfavored environments and negative under favored environments (Figure 6). This indicated the importance of panicle length as a characteristic for increasing yield under harsh environments. This positive relationship suggests that it is possible to improve yield by breeding cultivars with higher panicle length. These findings are supported by those obtained by Laza et al. [49,52], who highlighted the importance of panicle size as a promising characteristic for enhancing cultivars’ yield potential.

Considering our results, Giza179 and Giza178 exhibited high performance. Giza178 was the ideal genotype and had the widest adaptability. GZ1968-S-5-4 displayed the best genotype performance under Alexandria environments, where salinity stress affected the soil. At the same time, N22 presented the best values in Kharga oasis since high temperature coupled with low humidity are the common environmental conditions at this location.

## 4. Materials and Methods

### 4.1. Plant Materials

In the present research, a total of 34 rice genotypes were carefully selected. These genotypes could be classified into two sets. The first set consisted of elite Egyptian rice cultivars, specifically selected for their adaptability within the Egyptian rice production systems. These cultivars have been extensively cultivated and grown throughout Egypt, making them representative of the local rice landscape. The second set comprised international donor genotypes, known for their tolerance to salinity and water shortage stresses (Table 2). These donor genotypes were incorporated into the breeding program to enhance the resilience of the Egyptian elite cultivars against these specific constraints.

### 4.2. Experimental Locations and Climatic Conditions

The research trials were conducted across four distinct locations in Egypt, as presented in Table 3.

The trials were carried out over two consecutive growing seasons, 2021 and 2022. The temperature data for the four locations over the two years were acquired from https://weather.com/ (accessed on 3 December 2022), while the humidity data were extracted from the NASA POWER project for agricultural needs (https://power.larc.nasa.gov/ (accessed on 1 June 2023)).

Each of the four locations exhibits unique climatic conditions and soil properties. Kharga location, for instance, experiences high temperatures accompanied by very low humidity during the two growing seasons (Figure 7 and Figure 8, Appendix A). The climatic conditions varied between the two seasons (Figure 1 and Figure 2). Alexandria (Sobahya) location recorded the lowest maximum temperature and the highest humidity among the four locations, whereas the minimum temperature was exhibited at Gemmiza location during the two cropping seasons.

The physical and chemical analyses of the 8 environments’ soils are presented in Appendix A. The soils of the Gemmiza and Sakha locations are clay for the two seasons, while in Alexandria location was sandy clay loam. In contrast, Kharga oasis included loam sandy and clay sandy for 2021 and 2022 seasons, respectively. The Alexandria location soil analysis indicated the existence of high salt stress since Na^+^ was 59.10 and 58.94, whereas Cl^−^ amounted to 74.5 and 70.78 for the two seasons, 2021 and 2022, respectively.

### 4.3. Experimental Design and Data Recording

Randomized complete block design (RCBD) with three replications was adopted at each of the four different locations during the two successive seasons. The experiment consisted of total eight environments, representing the combination of the four locations and the two years. Each genotype’s seeds were planted in seedbed in May and transplanted after 25 days. The experimental plots were structured with five rows 2 m long with planting space of 20 cm × 20 cm. This layout was standardized for each genotype within every environment. To maintain the uniformity of all factors, except for the varying climatic conditions specific to each environment, standard agronomical practices such as field preparation, fertilizer application, and weed control were implemented. At the harvest stage, the plots were harvested at harvest stage and grain yield (g m^−1^) was recorded for each experimental plot after adjusting the moisture content. Furthermore, the yield-related characteristics, such as number of panicles per plant, panicle length (cm), thousand grain weight (g), and sterility (%), were studied.

### 4.4. Statistical Analysis

The statical packages available in R (R Core Team [53]) version 4.2.3. were used to conduct multivariate procedures for stability analysis according to AMMI and GGE. The “metan” package [20] was applied for AMMI analysis, whereas AMMI Model merged ANOVA and PCA techniques. The package was also employed for GGE biplot analysis [54], stability statistical analysis, and weighted average of absolute scores (WAAS) [43], while GGE-Biplot-GUI package was functioned to support the GGE-biplot-based analysis. Additionally, hierarchical cluster analysis was conducted via “Nbclust” package [55]. The AMMI analysis followed the below mathematical equation: Y_ge_ = μ + α_g_ + β_e_ + Σ_n_λ_n_γ_gn_δ_en_ + ρ_ge_, where Y_ge_ represents the grain yield for a particular genotype (g) in a given environment (e), μ is the genotype grand mean, α_g_ is the deviation of genotype performance from the mean, β_e_ stands for the deviation of environment from the mean, λ_n_ signifies the singular value of n component, γ_gn_ indicates the value of eigenvector for genotype (g), and δ_en_ is the value of eigenvector for e and ρ_ge,_ which is the remaining residual [55], while the GGE biplot model mathematical equation was P_ij_ = (y_ij_ − μ − δ_j_)/λ_j_ = (β_i_ + ϵ_ij_)/λ_j_, where P_ij_ is the matrix for genotype i and environment j, μ is the genotype grand mean, δ_j_ is the column (environment) main effect, λ_j_ is an evaluating factor, β_i_ is the row (genotype) main effect, and ϵ_ij_ stands for GEI, and y_ij_ is a two-way table for G and E [56]. The Spearman correlation coefficient was calculated among the grain yield, the studied yield-related characteristics using the GGaly R software package.

## 5. Conclusions

Different sets of genotypes showed high yield performance under different environments. While the environments had the highest portion of yield sum of squared deviations, the results showed that the selection should be conducted based on genotypes’ performance with respect to GEI. Furthermore, the results of AMMI, GGE biplot, and which–won–where showed that Giza178 had less response to the differences among the tested environments. This indicates that, compared to the tested genotypes, it has wide adaptability and capability as a climate-change-resilient genotype.

## Figures and Tables

**Figure 1 plants-13-00074-f001:**
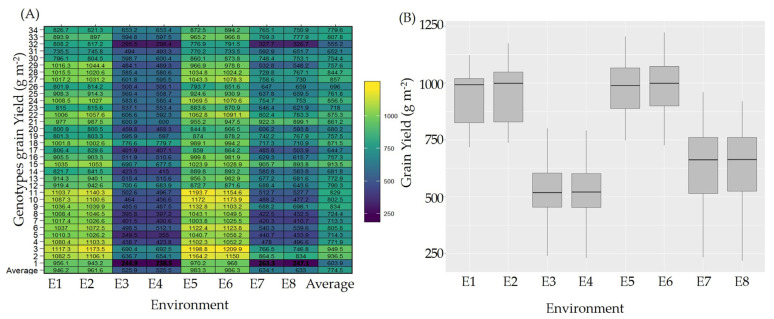
Genotypic performance across the different locations. (**A**) Genotypes’ grain yield means across the 8 different environments. (**B**) Box plot of grain yield for the 8 different environments explaining the differences among the 4 locations.

**Figure 2 plants-13-00074-f002:**
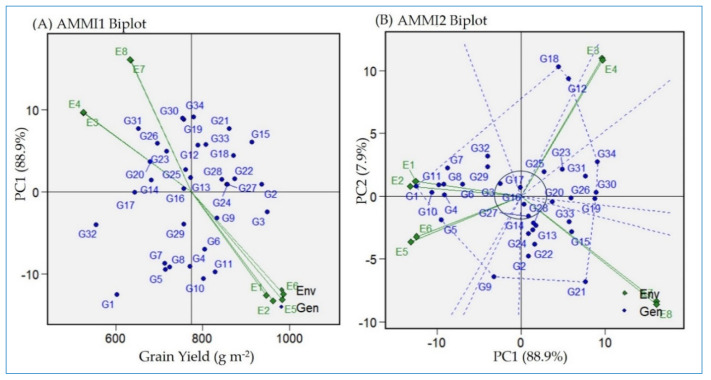
(**A**) The “AMMI1” biplot displays the main effect (GY) and IPC1 effect values explaining the relationship among tested genotypes and environments. (**B**) The “AMMI2” biplot displays the main axes of G+GEI effect (IPCA1 and IPCA2) values for the tested genotypes and environments. The tested genotypes are 34 (G1:G34 in blue color) grown in four locations in the two consecutive years, 2021 and 2022 (E1 and E2 = Sakha; E3 and E4 = Alexandria; E5 and E6 = Gemmiza; E7 and E8 = Kharga oasis).

**Figure 3 plants-13-00074-f003:**
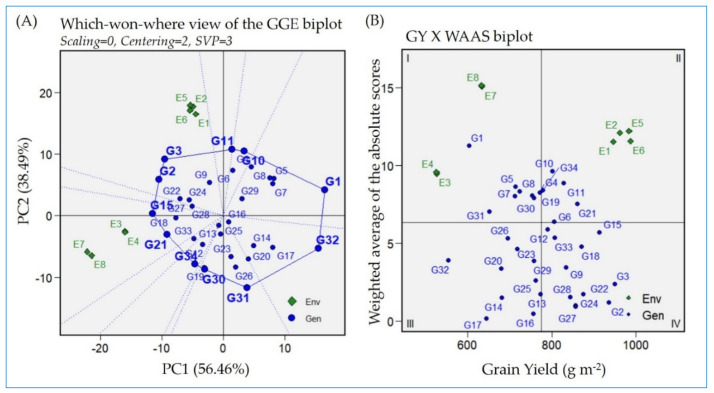
(**A**) The “which–won–where” polygon biplot displays the winning genotypes at each environment. (**B**) The “GY vs. WAAS” biplot displays the most-adopted genotypes across the tested environments. The tested genotypes are 34 (G1:G34 in blue color) grown in four locations in the two consecutive years, 2021 and 2022 (E1 and E2 = Sakha; E3 and E4 = Alexandria; E5, and E6 = Gemmiza; E7 and E8 = Kharga oasis).

**Figure 4 plants-13-00074-f004:**
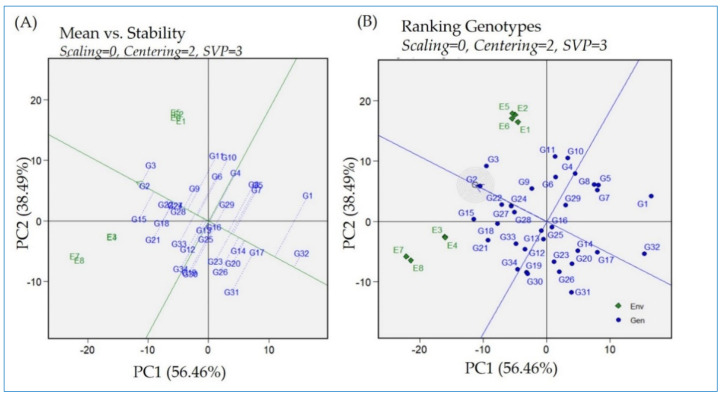
(**A**) The “mean versus stability” model describing the interaction effect of the tested rice genotypes evaluated across eight environments. (**B**) The “ranking genotypes” model of biplot to assess the ideal genotype. The tested genotypes are 34 (G1:G34 in blue color) grown in four locations in the two consecutive years, 2021 and 2022 (E1 and E2 = Sakha; E3 and E4 = Alexandria; E5 and E6 = Gemmiza; E7 and E8 = Kharga oasis).

**Figure 5 plants-13-00074-f005:**
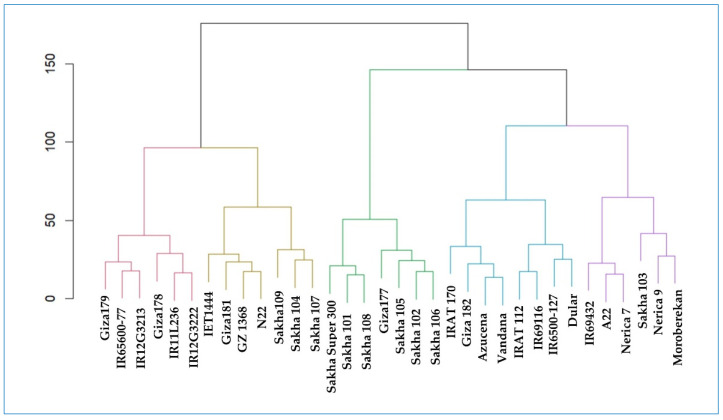
Hierarchical dendrogram classifying the 34 rice genotypes based on their ranks for GY and stability statistics conducted via Ward’s method.

**Figure 6 plants-13-00074-f006:**
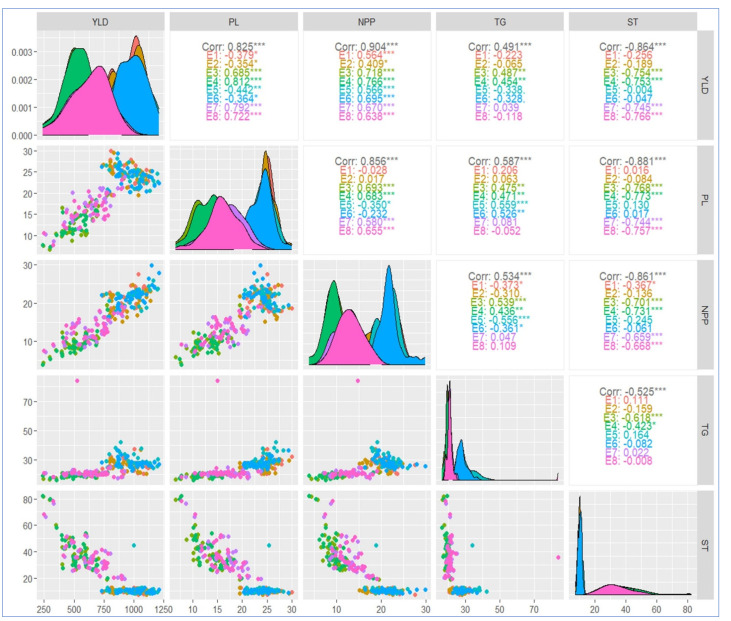
Spearman correlation coefficients for the yield, and other studied yield-related traits. Scatterplots of each trait’s pair of numeric variables are situated in the left part of the figure. Variable distribution is drawn on the diagonal. YLD: yield; PL: panicle length (cm); NPP: number of panicles per plant; TG: thousand grain weight (g); and ST: sterility (%). These traits were recorded for the tested genotypes grown in the four locations for the two consecutive years, 2021 and 2022 (E1 and E2 = Sakha; E3 and E4 = Alexandria; E5 and E6 = Gemmiza; E7 and E8 = Kharga oasis). * *p* ≤ 0.05, ** *p* ≤ 0.01, *** *p* ≤ 0.001.

**Figure 7 plants-13-00074-f007:**
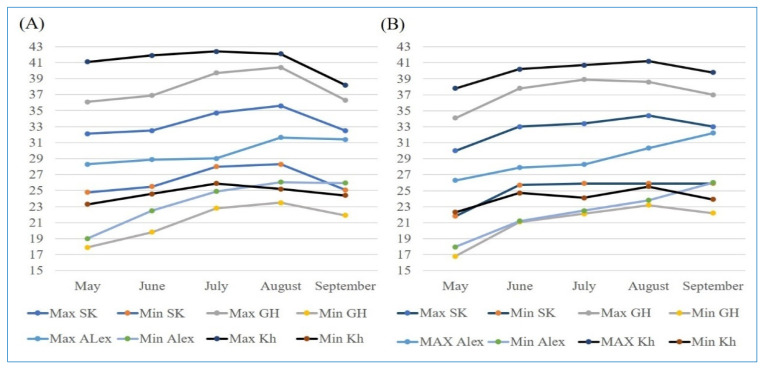
The monthly average maximum and minimum temperature (°C) at four locations during 2021 (**A**) and 2022 (**B**) rice seasons.

**Figure 8 plants-13-00074-f008:**
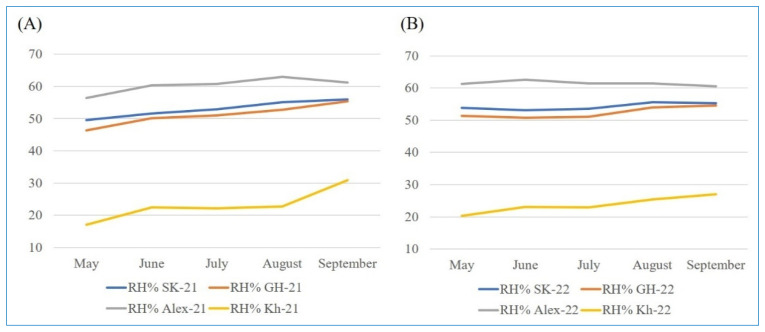
The monthly average of relative humidity (%) at the four locations during 2021 (**A**) and 2022 (**B**) rice seasons.

**Table 2 plants-13-00074-t002:** List of genotypes used in the study.

No.	Genotype	Origin	Type
1	Giza 177	Egypt	Japonica
2	Giza 178	Egypt	Indica/Japonica
3	Giza 179	Egypt	Indica/Japonica
4	Sakha 101	Egypt	Japonica
5	Sakha 102	Egypt	Japonica
6	Sakha 104	Egypt	Japonica
7	Sakha 105	Egypt	Japonica
8	Sakha 106	Egypt	Japonica
9	Sakha 107	Egypt	Japonica
10	Sakha 108	Egypt	Japonica
11	Sakha Super 300	Egypt	Japonica
12	IRAT 170	Ivory Cost	Tropical japonica
13	A22	Srilanka	Indica
14	Nerica 9	Ivory cost	Indica
15	IET 1444	Inbdia	Indica
16	Nerica 7	Ivory Cost	Indica
17	Moroberekan	Guinea	Japonica
18	GZ 1368-S-5-4	Egypt	Indica
19	Azucena	Philippine	Japonica
20	IRAT 112	Ivory Cost	Indica
21	N22	India	Aus
22	IR65600-77	Philippines	Indica
23	IR69116	Philippines	Indica
24	IR12G3213	Philippines	Indica
25	IR69432	Philippines	Indica
26	IR6500-127	Philippines	Indica
27	IR11L236	Philippines	Indica
28	IR12G3222	Philippines	Indica
29	Sakha 109	Egypt	Japonica
30	Vandana	India	Indica
31	Dular	India	Indica
32	Sakha 103	Egypt	Japonica
33	Giza 181	Egypt	Indica
34	Giza 182	Egypt	Indica

**Table 3 plants-13-00074-t003:** List of four locations used in the present study.

No.	Location (Governorate)	Altitude–Latitude
1	Sakha (Kafr El-Sheikh)	31.09° N and 30.9° E
2	Gemmiza (ElGharbya)	30.88° N and 31.05° E
3	Sobahya (Alexandria)	31.2° N and 29.9° E
4	Kharga oasis (New Valley)	25.4° N and 30.5° E

## Data Availability

The data supporting the findings of the current study are available within the paper and within its Appendix A published online.

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
