# Peer review of "Exploring Genetics by Environment Interactions in Some Rice Genotypes across Varied Environmental Conditions"

_plants, 2023, doi:10.3390/plants13010074_

Round 1

Reviewer 1 Report

Comments and Suggestions for Authors

The paper titled "Exploring Genetic by Environment Interactions in Some Rice Genotypes Across Varied Environmental Conditions" presents  a critical aspect of agricultural research, focusing on the development of rice genotypes resilient to the ever-increasing challenges posed by climate change. They undertook the evaluation of 34 rice genotypes across multiple locations with diverse environmental characteristics. Using advanced statistical techniques, such as additive main effects and multiplicative interactions (AMMI) modeling and genotype by environment (GGE) modeling, the research seeks to unravel the relationship between genotypes and environments, providing valuable insights into the stability and adaptability of various rice genotypes. The findings promise to contribute significantly to the development of climate-resilient rice varieties. However, there are some minor corrections need to be addressed before it can be accepted. 

Minor comments:
1. The first line of the abstract should be revised by mentioning the rationale  and the research problem of the work. 

2. Lines 41-90: It is essential to clarify the significance of "stability" and "adaptability" in rice breeding. Explain why these traits are crucial for addressing the challenges posed by climate change.

3. The introduction mentions "METs," but the acronym is not expanded. It would be helpful to spell out "Multi-Environment Trials (METs)" upon first use.

4. In line 74, clarify what "GGE polygon view" means for readers who may not be familiar with the term.

5. In lines 115-121, you discuss the four significant principal components of interaction (IPCAs). Consider explaining what these components represent and why they are significant in the context of the study.

6. Explain the significance of the GGE biplot and the "which-won-where" approach in lines 156-157 for readers who may not be familiar with these terms.

7. Interpret the results of the GY × WAAS biplot, especially regarding the genotypes located in different quadrants. Explain why some genotypes are considered low yielding and unstable, while others are stable and high yielding.

8. When discussing the hierarchical cluster analysis results, explain the criteria or indices used for grouping the genotypes. Provide a brief overview of the stability indices and their relevance to genotype grouping.

Comments on the Quality of English Language

Moderate editing of English language required

Author Response

Minor comments:
1. The first line of the abstract should be revised by mentioning the rationale and the research problem of the work. 

  • First, Authors would like to thank the respected reviewer for the valuable comments, which will enhance the quality of our research manuscript.
  • Second, the first line of the abstract changed to cope with the reviewer request.
  1. Lines 41-90: It is essential to clarify the significance of "stability" and "adaptability" in rice breeding. Explain why these traits are crucial for addressing the challenges posed by climate change.
  • Additional paragraph about stability and adaptability was added to the introduction section, lines 63-77.
  1. The introduction mentions "METs," but the acronym is not expanded. It would be helpful to spell out "Multi-Environment Trials (METs)" upon first use.
  • Multi-Environment Trials (METs) spelt out in lines 62-63, as requested accordingly.
  1. In line 74, clarify what "GGE polygon view" means for readers who may not be familiar with the term.
  • We clarified the meaning before the GGE polygon view (Line 91).
  1. In lines 115-121, you discuss the four significant principal components of interaction (IPCAs). Consider explaining what these components represent and why they are significant in the context of the study.
  • More details were added to clarify the meaning of iPCAs (L142-145).
  1. Explain the significance of the GGE biplot and the "which-won-where" approach in lines 156-157 for readers who may not be familiar with these terms.
  • Clarified as requested accordingly, L181-183.
  1. Interpret the results of the GY × WAAS biplot, especially regarding the genotypes located in different quadrants. Explain why some genotypes are considered low yielding and unstable, while others are stable and high yielding.
  • More clarification added to the section as requested accordingly, L215-233.
  1. When discussing the hierarchical cluster analysis results, explain the criteria or indices used for grouping the genotypes. Provide a brief overview of the stability indices and their relevance to genotype grouping.
  • The stability indices are existing in Table S5 (supplementary file). However, we would like to withdraw the review attention that we have several indices, each of them has a different methodology to determine the stability. We followed the clustering approach to collect the power of all indices in ranking the genotypes. Some stability indices were mentioned in L265-268.

Moderate editing of English language required

The manuscript has been revised and corrected accordingly.

Reviewer 2 Report

Comments and Suggestions for Authors

The authors submitted a manuscript entitled Exploring Genetic by Environment Interactions in Some Rice Genotypes Across Varied Environmental Conditions.

In their research, they looked at the influence of the environment on the influence of 34 varieties of rice. Although rice is one of the essential foods for human nutrition, I do not consider this research to this extent to be significant enough to be published in a journal with an impact factor.

The authors observed only four phenotypic characteristics of the varieties: panicle per plant, panicle length, 1000-grain weight and sterility. Although a large amount of statistical data is used in the study, it would be advisable to support it with another, more sophisticated method, e.g. gene expression.

The results achieved by the authors confirm only generally known information. Unfortunately, I do not see any innovative information in the submitted work that could be published.

Author Response

The authors submitted a manuscript entitled Exploring Genetic by Environment Interactions in Some Rice Genotypes Across Varied Environmental Conditions.

In their research, they looked at the influence of the environment on the influence of 34 varieties of rice. Although rice is one of the essential foods for human nutrition, I do not consider this research to this extent to be significant enough to be published in a journal with an impact factor.

The authors observed only four phenotypic characteristics of the varieties: panicle per plant, panicle length, 1000-grain weight and sterility. Although a large amount of statistical data is used in the study, it would be advisable to support it with another, more sophisticated method, e.g. gene expression. 

The results achieved by the authors confirm only generally known information. Unfortunately, I do not see any innovative information in the submitted work that could be published.

  • We respect the reviewer point of view; however, we would like to withdraw the attention towards the following:
  • There is a special need to estimate genotype by environment effect in the current years especially with different climate conditions. The approach used in our investigation mainly focusing on yield stability. However, we tried to have insights regarding other yield components that are reliable and have a significant impact on yield to conduct correlation analysis.
  • Several publications used the same approach and have been cited in our manuscript and published elsewhere in high impact factor peer reviewed journals such as;
  1. Mwiinga, B., Sibiya, J., Kondwakwenda, A., Musvosvi, C., & Chigeza, G. (2020). Genotype x environment interaction analysis of soybean (Glycine max (L.) Merrill) grain yield across production environments in Southern Africa. Field crops research256, 107922. https://www.sciencedirect.com/science/article/abs/pii/S0378429020312065
  2. Jiwuba, L., Danquah, A., Asante, I., Blay, E., Onyeka, J., Danquah, E., & Egesi, C. (2020). Genotype by environment interaction on resistance to cassava green mite associated traits and effects on yield performance of cassava genotypes in Nigeria. Frontiers in Plant Science11, 572200. https://www.frontiersin.org/articles/10.3389/fpls.2020.572200/full
  3. Huang, X.; Jang, S.; Kim, B.; Piao, Z.; Redona, E.; Koh, H.-J. Evaluating Genotype × Environment Interactions of Yield Traits and Adaptability in Rice Cultivars Grown under Temperate, Subtropical and Tropical Environments. Agriculture 2021, 11, 558. https://doi.org/10.3390/agriculture11060558
  4. Kebede, G.; Worku, W.; Feyissa, F.; Jifar, H. Genotype by Environment Interaction and Stability Analysis for Selection of Superior Fodder Yield Performing Oat (Avena Sativa L.) Genotypes Using GGE Biplot in Ethiopia. Genet. Genomics 2023, 28, 100192, https://doi.org/10.1016/j.egg.2023.100192.
  1. Kebede, G.; Worku, W.; Feyissa, F.; Jifar, H. Genotype by Environment Interaction and Stability Analysis for Selection of Superior Fodder Yield Performing Oat (Avena Sativa L.) Genotypes Using GGE Biplot in Ethiopia. Genet. Genomics 2023, 28, 100192, https://doi.org/10.1016/j.egg.2023.100192.

Reviewer 3 Report

Comments and Suggestions for Authors

Dear Author

I read the article Exploring Genetic by Environment Interactions in Some Rice
Genotypes Across Varied Environmental Conditions

please add the location on the table it will be good.

please add variance analysis. because you have to show the significant from hybrids and location after that do the AMMI analysis.

Author Response

Dear Author

I read the article Exploring Genetic by Environment Interactions in Some Rice
Genotypes Across Varied Environmental Conditions

please add the location on the table it will be good.

  • Locations and their Altitude-latitude were presented in table 3 (L468).

please add variance analysis. because you have to show the significant from hybrids and location after that do the AMMI analysis.

  • AMMI method integrates analysis of variance (ANOVA) and principal component analysis (PCA) into a unified approach that can be used to analyse multi-location trials. Besides, AMMI uses analysis of variance to study the main effects of genotypes and environments and a principal component analysis for the residual multiplicative interaction among genotypes and environments. In our study, the AMMI analysis performed by R package and table 1 is the results coming from this analysis, so this package performs the variance but did not show it in the resulted table.

Round 2

Reviewer 2 Report

Comments and Suggestions for Authors

I accept the references you send referring to similar outputs. I understand that your intention was to evaluate the yield parameters. It should be noted, however, that the breeding and selection of varieties based on only one parameter leads to a reduction in genetic diversity and may even lead to the loss of important genes. Therefore, I think it is not appropriate to evaluate the variety based only on morphological data. As part of your research, a large amount of data has been unambiguously evaluated, which shows and static evaluations.

After much consideration, I therefore accept the manuscript submitted by you.